# Diversity of Tick-Borne Pathogens in Tick Larvae Feeding on Breeding Birds in France

**DOI:** 10.3390/pathogens11080946

**Published:** 2022-08-20

**Authors:** Amalia Rataud, Clemence Galon, Laure Bournez, Pierre-Yves Henry, Maud Marsot, Sara Moutailler

**Affiliations:** 1Laboratory for Animal Health, Epidemiology Unit, Université Paris Est, ANSES, 94700 Maisons-Alfort, France; 2ANSES, INRAE, Ecole Nationale Vétérinaire d’Alfort, UMR BIPAR, Laboratoire de Santé Animale, 94700 Maisons-Alfort, France; 3ANSES, Nancy Laboratory for Rabies and Wildlife, 54220 Malzéville, France; 4Mécanismes Adaptatifs et Evolution (MECADEV UMR 7179), Muséum National d’Histoire Naturelle, CNRS, 91800 Brunoy, France; 5Centre de Recherches sur la Biologie des Populations d’Oiseaux (CRBPO), Centre d’Ecologie et des Sciences de la Conservation (CESCO UMR 7204), Muséum National d’Histoire Naturelle, CNRS, 75005 Paris, France

**Keywords:** wild bird, breeding season, tick, tick-borne pathogen

## Abstract

Birds play a role in maintaining tick-borne diseases by contributing to the multiplication of ticks and pathogens on a local scale during the breeding season. In the present study, we describe the diversity of tick and pathogen species of medical and veterinary importance in Europe hosted by 1040 captured birds (56 species) during their breeding season in France. Of the 3114 ticks collected, *Ixodes ricinus* was the most prevalent species (89.5%), followed by *I. frontalis* (0.8%), *I. arboricola* (0.7%), *Haemaphysalis concinna* (0.5%), *H. punctata* (0.5%), *Hyalomma* spp. (0.2%), and *Rhipicephalus* spp. (0.06%). Because they may be representative of the bird infection status for some pathogen species, 1106 engorged tick larvae were screened for pathogens. *Borrelia burgdorferi* sensu lato was the most prevalent pathogen genus in bird-feeding larvae (11.7%), followed by *Rickettsia* spp. (7.4%), *Anaplasma* spp. (5.7%), *Babesia* spp. (2.3%), *Ehrlichia* spp. (1.4%), and *B. miyamotoi* (1%). *Turdidae* birds (*Turdus merula* and *T. philomelos*), *Troglodytes troglodytes*, and *Anthus trivialis* had a significantly higher prevalence of *B. burgdorferi* s.l.-infected larvae than other pathogen genera. This suggests that these bird species could act as reservoir hosts for *B. burgdorferi* s.l. during their breeding season, and thus play an important role in acarological risk.

## 1. Introduction

Ticks are the second most important disease vector worldwide in human health, after mosquitoes, and the first in animal health [1]. They are obligate parasites, and their three life stages (larva, nymph and adult) can have their own trophic preferences [2,3]. Ticks can host and transmit a wide variety of pathogens, whether bacteria, viruses or parasites, to a broad spectrum of vertebrate hosts [3]. A better knowledge of the diversity of pathogens that ticks can host and transmit, their different modes of transmission, and the roles of tick hosts in the epidemiological cycle of pathogens is essential to improve the control of tick-borne diseases and reduce the acarological risk [3]. Hosts can participate in tick-borne pathogen dynamics by feeding ticks, thereby allowing them to evolve to their next life stage, and/or by transmitting pathogens to ticks if they are competent reservoir hosts [4]. Birds are important hosts to consider as they can disseminate ticks and their associated pathogens on a large scale during their migration period before and after breeding [5,6,7]. Birds can also participate in the local dynamics of ticks and pathogens during their sedentary periods by feeding ticks (and infecting them or being infected by them) during breeding [8] or wintering [9,10]. In Europe, birds can contribute to the population dynamics of a wide variety of tick species with potential medical and veterinary importance, including *Ixodes ricinus*, *Haemaphysalis concinna*, *H. punctata*, *Hyalomma marginatum,* and *Hy. lusitanicum* [11,12,13,14,15,16,17,18]. Birds also feed specialist, ornithophilic ticks such as *I. arboricola*, *I. lividus*, *I. frontalis*, *I. festai* and *I. eldaricus* [11,12,15,17,19], some of which also transmit pathogens [20].

Through ticks, birds can participate in the dynamics of pathogens of medical or veterinary importance, whether bacteria (*Anaplasma* spp., *Borrelia* spp., *Ehrlichia* spp., *Rickettsia* spp., *Coxiella* spp.) [14,18,21,22,23,24,25,26,27,28], parasites (*Babesia* spp.) [22,23], or viruses (tick-borne encephalitis virus, Crimean-Congo haemorrhagic fever) [29,30,31].

The objective of this preliminary study was to carry out an inventory of tick and pathogen species (concentrating on bacteria and parasites) of medical and veterinary importance in Europe, that are hosted by common birds in France during their breeding season. This recurring annual event in the life cycle of birds, in spring, is important to consider as it overlaps with the peak activity of ticks (*I. ricinus*) and the period during which humans do outdoor activities and are more exposed to infectious tick bites. From this inventory, we sought to calculate tick infection rates and prevalence among birds for the pathogen genera identified in order to reveal any evidence of a higher level of infection of certain bird species by specific pathogens.

We therefore evaluated the diversity of tick species hosted by a wide variety of bird species and screened for a broad spectrum of pathogens (27 species from five genera of bacteria, eight species from two genera of parasites) harboured by bird-feeding tick larvae. While co-feeding is negligible, engorged larvae can be considered as an indicator of the infection status of birds for pathogens with negligible transovarial transmission, such as *Borrelia burgdorferi* sensu lato (*Bbsl*) and *A. phagocytophilum* [32,33]. For other pathogens, engorged larvae are considered as proxies of prevalence among birds and transovarial transmission. From the literature, we hypothesised (i) that *I. ricinus* would be most abundant in the sample, since it is the most common tick in Europe, with an activity peak in the spring [34], (ii) that *Borrelia* spp. and *Rickettsia* spp. would be the most frequently detected tick-borne pathogens (TBPs) as they are the most prevalent in bird-feeding ticks [14,23,24,28,35,36], and (iii) that birds belonging to the *Turdidae* family and *Parus major* would host a higher proportion of infected larvae, in particular for specific pathogen species such as *Bbsl* [12,14,28,37].

## 2. Results

### 2.1. Bird Capture and Tick Collection

Ticks were collected from a total of 1040 birds belonging to 56 species (491 birds in 2019, 549 birds in 2020) captured from March to September (5% of bird captures before 16 May, 50% before 12 June and 95% before 10 July). *Erithacus rubecula*, *Sylvia atricapilla*, *Turdus merula* and *Parus major* were the most commonly caught (more than 50%). The 3114 ticks collected belonged to five species, *I. ricinus* being the most frequently found (89.5% of collected ticks, n = 2787; Table 1 and Appendix A). A minor fraction (7.9%) of these ticks could not be identified: 0.4% (n = 14) were identified at genus level only (four *Ixodes* spp., three *Haemaphysalis* spp., five *Hyalomma* spp. and two *Rhipicephalus* spp.) and 7.5% (n = 233) were too damaged for morphological identification. Molecular techniques failed to confirm identification. As *I. ricinus* ticks were detected throughout France, Appendix A represents the geographical distribution respectively of *Ixodes* ticks other than *I. ricinus* (*I. arboricola* and *I. frontalis*), *Haemaphysalis* spp. (*H. concinna* and *H. punctata*), *Hyalomma* spp. and *Rhipicephalus* spp. Among the ticks collected, 55.1% (n = 1715) were nymphs, 41.3% (n = 1285) were larvae—of which 86.1% (n = 1106) were engorged—1.9% (n = 60) were adult females, 0.03% (n = 1) were adult males and 1.7% (n = 53) were too damaged for morphological life-stage identification (Table 1). However, it should be noted that the collection protocol was favourable to nymphs and engorged larvae, so these proportions should not be considered as representative of the actual age structure of ticks feeding on birds.

### 2.2. Tick-Borne Pathogen Infection Rates in Engorged Larvae and Prevalence among Birds

TBPs were detected from the 1106 engorged larvae that were collected from 442 birds belonging to 36 species. *Bbsl* was the most common pathogen genus found, with a larva infection rate (i.e., the number of TBP-positive engorged larvae out of the total number of engorged larvae collected from birds) of 11.7%. Its prevalence among birds (i.e., the number of tick-infested birds with at least one TBP-positive larva out of the total number of sampled birds) was of 15.8% (Figure 1A, Table 2 and Appendix A). *Rickettsia* spp. (larva infection rate = 7.4%, prevalence = 13.3%) was the second most common pathogen genus detected, followed by *Anaplasma* spp. (larva infection rate = 5.7%, prevalence = 10.2%), *Babesia* spp. (larva infection rate = 2.3%, prevalence = 4.1%), *Ehrlichia* spp. (larva infection rate = 1.4%, prevalence = 2.7%) and *B. miyamotoi* (larva infection rate = 1%, prevalence = 2.3%) (Figure 1A, Table 2 and Appendix A). *Bartonella* spp., *Coxiella* spp., *Francisella* spp. and *Theileria* spp. were not detected in any engorged larvae. Some pathogen species could not be clearly identified because DNA sequencing failed and did not allow precise identification between several pathogen species (this was the case for 23 larvae positive for *Rickettsia* spp., 22 larvae positive for *Babesia* spp., nine larvae positive for *Ehrlichia* spp. and one larva positive for *Borrelia* spp., Table 2). The accession numbers of the sequences submitted for tick and TBP species are presented in (Appendix A).

Pooling all pathogen genera, prevalence among birds differed between bird species and was significantly higher in *T. merula* (prevalence = 76.7%) and *T. philomelos* (prevalence = 80%) than in *E. rubecula* (prevalence = 30.8%, Figure 1B). Pooling all bird species, prevalence among birds also differed according to the pathogen genus and was significantly lower in *Anaplasma* spp. (prevalence = 10.2%), *Babesia* spp. (prevalence = 4.1%), *Ehrlichia* spp. (prevalence = 2.7%) and *B. miyamotoi* (prevalence = 2.3%) than in *Bbsl* (prevalence = 15.8%). There was no significant difference in the prevalence between *Bbsl* and *Rickettsia* spp. (prevalence = 13.3%, Figure 1A). Finally, the prevalence among birds for *Bbsl* differed among bird species: *T. merula* (prevalence = 67.4%), *T. philomelos* (prevalence = 60%), *Troglodytes troglodytes* (prevalence = 29.4%) and *Anthus trivialis* (prevalence = 50%) were significantly more infected with *Bbsl* than *E. rubecula* (prevalence = 4.7%, Figure 1C). There was no significant difference in prevalence according to bird species for *Anaplasma* spp., *Babesia* spp., *Ehrlichia* spp., *Rickettsia* spp., and *B. miyamotoi*. Pooling all pathogen genera, prevalence among birds was higher in 2020 than in 2019.

Co-infections of two pathogen genera were detected in 2.9% (n = 32) of the engorged larvae, the most prevalent pathogens (*Borrelia* spp., *Rickettsia* spp. and *Anaplasma* spp.) being represented the most. Moreover, 0.5% of engorged larvae (n = 6) were co-infected with three pathogen genera (Appendix A).

## 3. Discussion

This study characterised the diversity of ticks (five species from two genera plus two species identified at genus level only) and TBPs (13 species from five genera) of veterinary and medical importance hosted by 56 species of European wild birds during their breeding season in spring, in a temperate region (France). As immature ticks of the species collected (*I. ricinus*, *I. frontalis*, *I. arboricola*, *H. concinna*, *H. punctata*) mostly feed on small mammals and birds [9,15,17,38,39,40] and their activity peaks during the breeding season of birds [9,34,40,41,42], we mostly collected nymphs and larvae. Pathogens were not equally represented in tick larvae from birds, *Bbsl* and *Rickettsia* spp. prevailing. *Bbsl* was more prevalent in engorged larvae collected from certain bird species (*T. merula*, *T. philomelos*, *T. troglodytes*, *A. trivialis*), whereas *Rickettsia* spp. appeared to be equally represented among host bird species. Bird prevalence by all pathogen genera was higher in 2020 than in 2019, a long-term study should be conducted to test whether the infection status of birds varies over time.

As expected, we found that *I. ricinus* was the predominant tick species hosted by birds, representing 89.5% of all ticks collected. All the life stages of this species were collected from birds, and it was found on 49 out of 56 bird species. As a generalist tick, *I. ricinus* can carry a wide range of TBPs, such as *Bbsl* [43], *Anaplasma* spp. [44], *Rickettsia* spp. [45], *Babesia* spp. [46], *B. miyamotoi* [47] and *Ehrlichia* spp. [48] as was found in our study, but also *Francisella* spp. [49] and *Coxiella* spp. [50] which are occasionally found in bird-feeding ticks [35,51,52]. Only a few other tick species were collected from birds in this study. This was expected for the ornithophilic tick *I. frontalis*, since nymph and adult *I. frontalis* are sporadically present on the ground throughout the year, while larvae activity peaks in autumn and decreases in winter [9]. Like other studies, we found *I. frontalis* to be infected by *Bbsl* [9], but no *A. phagocytophilum* contrary to Agoulon et al. [9]. This may be due to the small sample size of *I. frontalis* engorged larvae (n = 2). Moreover, a few individuals belonging to the ornithophilic and nidicolous tick species *I. arboricola* [17] were collected. As expected [17], all *I. arboricola* (whatever their life stage) were found on cavity-nesting bird species: three tit species (*P. major*, *Poecile palustris*, *Cyanistes caeruleus*) and an owl (*Athene noctua*). No engorged larvae were found, so we could not screen *I. arboricola* for pathogens, but this tick is known to bear the two most prevalent pathogens, *Rickettsia* spp. and *Bbsl* [20,53,54].

Three genera other than *Ixodes* spp. were found on the collected birds: *Haemaphysalis* spp., *Hyalomma* spp. and *Rhipicephalus* spp. A few individuals (larvae, nymphs) of two tick species belonging to the genus *Haemaphysalis* spp. (*H. concinna* and *H. punctata*) were collected from birds. *H. concinna* is common in deciduous or mixed forests near the shores of lakes or rivers in Europe and Asia [39]. This could explain the very small number of individuals collected in this study compared to other species like *I. ricinus*, as this is not the preferred environment for bird capture. In Central Europe, the peak activity of all the life stages of *H. concinna* overlaps with the bird breeding season [42]. The absence of adult *H. concinna* found in our study could be explained by the fact that this life stage mostly feeds on deer and farm animals [15]. As found in other studies, *H. concinna* larvae were infected by *Rickettsia* spp. [39]. However, we did not find *Bbsl*, *Coxiella* spp., *Francisella* spp. or *Babesia* spp. [39], possibly due to the small sample size. One *H. punctata* nymph and a few larvae were collected in our study. The very small number of nymphs and the absence of adults could be explained by the fact that these life stages mainly feed on wild ungulates, domestic animals and medium-sized mammals [40]. As found in other studies, *H. punctata* larvae were infected by *Anaplasma* spp. and *Rickettsia* spp. [55,56], but not by *Babesia* spp. or *Bbsl* contrary to Phipps et al. [57]. Finally, five nymphs belonging to *Hyalomma* spp. were collected from *Acrocephalus scirpaceus*, a finding already reported in the literature [58], and two adults belonging to *Rhipicephalus* spp. were collected from *Aquila fasciata* as reported in [59] for *R. bursa*.

*Turdus merula* and *T. philomelos* were the most infected bird species for all pathogen genera considered. Species belonging to the *Turdidae* family have been shown to play an important role in TBP circulation [14,28,37]. Bacteria belonging to *Bbsl* were the most prevalent TBPs (prevalence among birds = 15.8%, larva infection rate = 11.7%) as is the case in many previous studies [14,23,24,28,35,36]. The larva infection rate obtained in our study was similar to that found in a study conducted in Italy (11%; [35]) and was lower than larva infection rates found by three studies conducted in Europe, respectively in Switzerland (15.1%; [28]), the Netherlands and Belgium (19.5%; [14]) and in 11 European countries (20%; [60]). It was higher than that found in Latvia (3%; [23]) and Norway (0% in spring; [36]) perhaps due to the smaller sample tested in these studies (respectively 37 and 52 tested larvae). *B. garinii* was the most prevalent species (prevalence among birds = 12.2%, larva infection rate = 9%), and is already known to be associated with birds [23,34,61]. Apart from *B. garinii*, other *Bbsl* species associated with birds found in our study—*B. valaisiana* (prevalence = 3.4%, larva infection rate = 1.8%) and *B. turdi* (prevalence = 0.7%, larva infection rate = 0.4%)—were already known to circulate and multiply mainly in bird hosts [23,34,62]. This last species was, however, nearly as rare as the generalist TBP *Borrelia burgdorferi* sensu stricto (prevalence = 0.5%, larva infection rate = 0.2%) and the rodent-associated *B. afzelii* (prevalence = 0.5%, larva infection rate = 0.2%) [63]. It thus appears that we mainly have a community of bacteria belonging to *Bbsl* essentially linked to bird host communities. The high prevalence of *B. garinii* among birds could confirm the ability of birds to act as a reservoir for this pathogen [64], taking into account the fact that co-feeding and transovarial transmission could occur sporadically [32,33,65,66]. *Bbsl* prevalence among birds differed significantly between bird species, with *T. merula*, *T. philomelos*, *T. troglodytes*, and *A. trivialis* being the most infected. These bird species are known to actively participate in the circulation of *Bbsl* [14,28,37,67,68]. Contrary to our hypothesis [12], *Bbsl* was not very prevalent among *P. major* specimens compared to other bird species. We may conclude that infection by *Bbsl* has a structuring effect according to the bird species, with some species appearing to be more involved in the circulation of this pathogen.

The second most prevalent TBP genus in larvae collected from birds was *Rickettsia* spp. (prevalence= 13.3%, larva infection rate = 7.4%). This larva infection rate is similar to rates found in Sweden (6.8%; [25]), Slovakia (5.8%; [21]), and Latvia (5%; [23]). *R. helvetica* was the most prevalent *Rickettsia* species (prevalence 8.1%, larva infection rate = 4.1%), and birds have already been identified as participating in its circulation [14,21,23] and acting as potential reservoir hosts [69], although *R. helvetica* can be transovarially transmitted [70,71]. Unexpectedly, *R. aeschlimannii* was found in four larvae belonging to *I. ricinus* collected from four birds (prevalence = 0.9%, larva infection rate = 0.4%), whereas it is usually hosted by *Hyalomma* spp. ticks [52,72,73]. This implies that *I. ricinus* (and its bird hosts) could contribute to *R. aeschlimannii* dynamics as suggested in Mancini et al. [74], where it was detected in a questing *I. ricinus* and in Wallménius et al. [22], where it was detected in *I. frontalis* ticks collected from birds. However, it could also suggest that only DNA traces of *R. aeschlimannii* were found in engorged larvae. Finally, *R. slovaca* has been sporadically found in birds (prevalence = 0.5%, larva infection rate = 0.9%). Unexpectedly, it was detected in one *I. ricinus* larva (and nine unidentified larvae), whereas it is usually hosted by *Dermacentor marginatus* and *D. reticulatus* in Europe, which are considered as the most important vectors [75]. This implies that *I. ricinus* ticks hosted by birds could participate in *R. slovaca* dynamics, as was suggested only once in Mărcuţan et al. [75], where it was detected in an *I. ricinus* collected on *T. merula*. *R. slovaca* can be transmitted transovarially from the female to the larvae in *Dermacentor* spp. [76]; to our knowledge no such evidence has been found for *I. ricinus*. The high larva infection rate for *Rickettsia* spp. could not suggest that birds are competent as reservoir hosts for this pathogen, as transovarial transmission could often occur. According to the statistical analysis, there was no structuring effect of *Rickettsia* spp. infection depending on bird species, which suggests that there is no particular bird species with a major role in the circulation of this pathogen among sampled bird species.

*Anaplasma* spp. was the third most prevalent TBP genus in larvae collected from birds, *A. phagocytophilum* being the only detected species (prevalence = 10.2%, larva infection rate = 5.7%). The larva infection rate in our study was similar to that found in the Netherlands and Belgium (4.6%; [14]) and higher than that found in Latvia (2.7%; [23]).This species has already been found in many ticks collected from birds [36,77,78], which play a role in the species’ circulation by feeding ticks, dispersing infected ticks and/or infecting ticks, as was demonstrated in Johnston et al. and Keesing et al. [79,80]. The relatively high larva infection rate for *A. phagocytophilum* could suggest that birds are competent as reservoir hosts for this pathogen, as transovarial transmission is negligible in *I. ricinus* ticks [33]. Moreover, according to our statistical analysis, there is no structuring effect of *Anaplasma* spp. infection depending on bird species, which suggests that there is no particular bird species with a major role in the circulation of this pathogen among sampled bird species.

Some larvae collected from birds were positive for *Babesia* spp. (prevalence = 4.1%, larva infection rate = 2.3%). The larva infection rate in our study was lower than that found in Latvia (5%; [23]). *B. venatorum* (prevalence = 0.7%, larva infection rate = 0.3%) was the only species detected. While the commonly known reservoirs of this species are large domestic and wild ruminants, including cattle and roe deer [81], birds have often been identified as being involved in *B. venatorum* circulation by hosting infected ticks [23,82,83]. 

Finally, *Ehrlichia* spp. was the least prevalent TBP genus in engorged larvae (prevalence = 2.7%, larva infection rate = 1.4%). This genus is not often detected in bird-feeding ticks in Europe [73,84]. *E. canis* was the most prevalent (prevalence = 0.2%, larva infection rate = 0.1%). The role of birds in the circulation of a species close to *E. canis* has already been identified in Brazil, where it was detected in the blood of *Coragyps atratus* [85], *Asio clamator* and *Rupornis magnirostris* [86]. Moreover, a species close to *E. chaffeensis* (prevalence =1.1%, larva infection rate = 0.5%) was also detected in larvae collected from birds in this study. The role of birds in the circulation of this species has already been demonstrated by Hornok et al. [27], who detected it in the blood of a *T. philomelos* specimen in Hungary, and Machado et al. [85], who detected it in the blood of a *Falcos sparverius* in Brazil. These results show that birds are involved in *Ehrlichia* spp. circulation at the very least by feeding potentially infected ticks.

Another *Borrelia* spp. species, *B. miyamotoi*, which does not belong to *Bbsl*, was detected in larvae feeding on birds (prevalence = 2.3%, larva infection rate = 1%). This species has already been shown to be associated with birds [14,28,36].

To conclude, this study revealed that despite hosting a relatively low diversity of tick species, birds participate in the circulation of a high diversity of TBP species (*B. garinii*, *A. phagocytophilum* and *R. helvetica* being the most prevalent) during their breeding season in France. The higher prevalence of the generalist tick species *I. ricinus* over more specialist ones could influence TBP circulation. Indeed, infected generalist ticks can increase the acarological risk because they can feed on a wide variety of hosts and thus be infected or infect them, contrary to more specialist ticks that feed on a restricted panel of hosts. Generalist ticks can transmit specialist TBPs to other hosts in the epidemiological system [87,88]. Moreover, birds may play a direct role in local TBP circulation by infecting ticks during their bloodmeal (reservoir-competent hosts), an indirect role by acting as a ‘bridge’ in co-feeding transmission (if they allow the aggregation and simultaneous feeding of ticks at multiple life stages), or an inconsequential role in the case of transovarial transmission. In the latter case, birds participate in TBP circulation by feeding infected ticks, thereby producing infected ticks in the next life stage [89]. Although birds belonging to the *Turdidae* family would appear to be more involved than others in the circulation of TBP genera (having a higher prevalence among birds than other species), further research is needed to determine which bird-related factors influence their contribution to the circulation of pathogenic species. Indeed, as reservoir hosts, birds may contribute differently than other hosts they live longer and offer a higher species diversity than other reservoir hosts such as rodents, making their role in TBP circulation important both by disseminating ticks over long distances during migration and by producing infected ticks on a local scale (acarological risk) during the breeding season. Further research on the bird compartment (less studied than that of mammals), and in particular on the reservoir host potential of avian species, which depends on their realized reservoir competence, tick production, and density in the environment [4], would clarify the qualitative and quantitative role of birds in the acarological risk of tick-borne diseases.

## 4. Materials and Methods

### 4.1. Bird Capture and Tick Collection

Birds were captured by authorised bird-ringers during the breeding season in 2019 and 2020 at 110 sites spread across France (Appendix A). The majority of birds (95%) were sampled at Constant ringing Effort Sites (i.e., fixed plot with a fixed monitoring design), where the three annual capture sessions take place one morning (6 am-12 noon) every two to four weeks between May and early July. The sampling plots cover two to four hectares, across which are spread between ten and twenty 12-metre by 2.5-metre mist nets positioned about 50 m apart. With this method, only birds flying between the ground and about three metres above are sampled. The other sampled birds (5%) were obtained using other bird monitoring designs and were included to increase the range of documented bird species. A maximum of ten ticks feeding on an individual bird was asked to be collected with tweezers (all over the bird’s body) and to be immersed in a single tube filled with ethanol (70%) by the ringers.

### 4.2. Morphological Tick Identification

Tick stage and species (when possible) were identified morphologically using binocular loupes [90,91]. Only engorged larvae were analysed by molecular methods in the present study, as they can be considered an indicator of bird infection status under the strong assumption of negligible transovarial transmission or co-feeding. Indeed, if co-feeding is negligible, engorged larvae can be considered to indicate the infection status of birds for pathogens with negligible transovarial transmission. For other pathogens, engorged larvae are considered as proxies of prevalence among birds and transovarial transmission. We verified the morphological identification of species of engorged larvae by PCR amplification targeting a fragment of the COI gene (see hereafter) when: (i) the larva was too damaged, (ii) the larva was positive for at least one pathogen genus, and (iii) the identified species was other than *I. ricinus*, or identified only at genus level.

### 4.3. DNA Extraction and Pre-Amplification

DNA was extracted from individual larvae using the Nucleopsin tissue kit (Macherey Nagel, Düren, Germany) according to the manufacturer’s instructions (as in Banović et al. [92]). To enhance the detection of pathogen DNA, total DNA was pre-amplified with the PreAmp Master Mix (Fluidigm, San Francisco, CA, USA) according to the manufacturer’s instructions as in Banović et al. [93].

### 4.4. Detection of Tick-Borne Pathogens: DNA Microfluidic Real-Time PCR

To detect the bacteria and parasites of medical and veterinary importance in Europe (27 bacteria species from five genera, eight parasite protozoa species from two genera), the BioMark™ real-time PCR system (Fluidigm, San Francisco, CA, USA) was used for high-throughput microfluidic real-time PCR amplification using the 48.48 dynamic arrays (Fluidigm, San Francisco, CA, USA) as in Banović et al. and Boularias et al. [93,94]. All the pathogens were confirmed by PCR or nested PCR as in Banović et al. and Boularias et al. [92,94] using the primers presented in Table 3. The PCR products were sequenced by Eurofins Genomics (Cologne, Germany), then assembled using the BioEdit software (Ibis Biosciences, Carlsbad, CA, USA). Our results were compared with the online BLAST (http://www.ncbi.nlm.nih.gov/blast, accessed on 8 April 2022) using the GenBank dataset (https://www.ncbi.nlm.nih.gov/, accessed on 8 April 2022) to identify the sequenced microorganisms. The accession numbers of the sequences submitted for tick and TBP species are given in (Appendix A).

### 4.5. Statistical Analyses

We calculated larva infection rates as the number of TBP-positive engorged larvae out of the total number of engorged larvae collected from sampled birds. We then calculated the prevalence among birds (i.e., the number of birds hosting ticks with at least one TBP-positive engorged larva out of the total number of sampled birds) for each TBP genus detected and each bird species. Next, we tested the existence of a structuring effect of bird infections by all pathogen genera according to bird species and pathogen genus. This entailed using a binomial generalized linear model to test the prevalence of pathogens among birds for all pathogen genera according to the bird species and the pathogen genus. We then used another generalized linear model to test the prevalence of each pathogen genus among birds separately according to the bird species. *Erithacus rubecula* was set as the reference bird species because it was the most represented, and *Bbsl* as the reference pathogen genus because it was found on the greatest number of birds. Finally, we tested the existence of an effect of year on the bird prevalence by all pathogen genera using a generalized linear model. We did not test the effect of tick species on the prevalence among birds because the major tick species was *I. ricinus* and the samples for other tick species collected were too small. Bird species represented by fewer than four collected birds were removed from these analyses to increase statistical power. Results were considered significant when *p* < 0.05.

## Figures and Tables

**Figure 1 pathogens-11-00946-f001:**
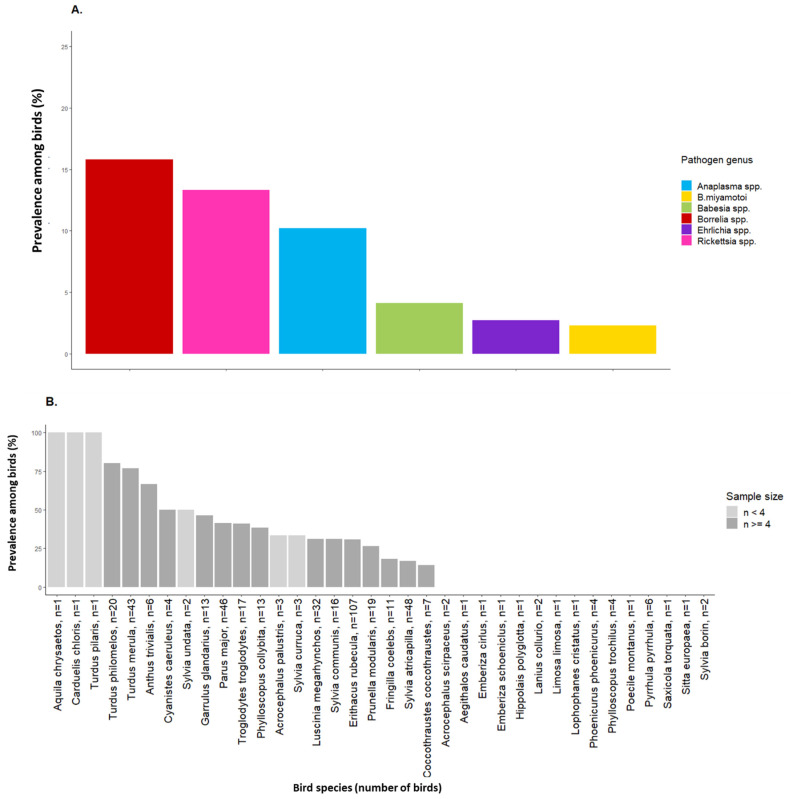
Prevalence among birds according to pathogen genus (**A**); prevalence according to bird species and sample size, pooling all pathogen genera (**B**); prevalence for the most prevalent pathogen genera (*Bbsl*, *Rickettsia* spp., *Anaplasma* spp.) according to bird species (**C**). The number of birds sampled is indicated after the bird species name for Figures (**B**,**C**). Bird species are ranked in decreasing order of pathogen prevalence among birds in Figure (**B**) and from the most frequently to the least frequently sampled bird species in Figure (**C**).

**Table 1 pathogens-11-00946-t001:** Number of ticks collected per tick species and tick life stage (percentage calculated out of all the ticks, n = 3114).

	Life Stage	Male	Female	Nymph	Larva (Engorged)	Unidentified	Total (%)
Species							
Genus *Ixodes*							
*I. ricinus*		0	29	1591	1167 (1039)	0	2787 (89.5%)
*I. frontalis*		0	16	8	2 (2)	0	26 (0.8%)
*I. arboricola*		0	3	17	2 (0)	0	22 (0.7%)
*I.* spp.		0	0	0	4 (2)	0	4 (0.1%)
Genus *Haemaphysalis*							
*H. concinna*		0	0	5	11 (11)	0	16 (0.5%)
*H. punctata*		0	0	1	15 (15)	0	16 (0.5%)
*H.* spp.		0	0	0	3 (3)	0	3 (0.1%)
Genus *Hyalomma*							
*H.* spp.		0	0	5	0	0	5 (0.2%)
Genus *Rhipicephalus*							
*R.* spp.		1	1	0	0	0	2 (0.06%)
Unidentified ^a^		0	11	88	81 (34)	53	233 (7.5%)
Total (%)		1 (0.03%)	60 (1.9%)	1715 (55.1%)	1285 (41.3%)	53 (1.7%)	3114

^a^ Unidentified because morphologically damaged.

**Table 2 pathogens-11-00946-t002:** Engorged larva infection rates (percentage of infected engorged larvae out of the total number of engorged larvae collected from sampled birds) per pathogen and engorged larva species. The number of infected engorged larvae is noted in brackets.

	Pathogen Species	*I. ricinus*	*I. frontalis*	*Ixodes* spp.	*H. concinna*	*H. punctata*	*Haemaphysalis* spp.	Unidentified	Number of Birds with TBP-Positive Larvae
Pathogen Species									
Genus *Anaplasma*									
*A. phagocytophilum*		5.8 (60)	0	0	0	6.7 (1)	0	5.9 (2)	45
Genus *Babesia*		2.3 (24)	0	0	0	0	0	2.9 (1)	18
*B. venatorum*		0.3 (3)	0	0	0	0	0	0	3
*B.* spp.		2 (21)	0	0	0	0	0	2.9 (1)	15
Genus *Bbsl ^a^*		11.9 (124)	100 (2)	50 (1)	0	0	33.3 (1)	2.9 (1)	70
*B. afzelii*		0.2 (2)	0	0	0	0	0	0	2
*Bbss ^b^*		0.2 (2)	0	0	0	0	0	0	2
*B. garinii*		9.2 (96)	50 (1)	50 (1)	0	0	33.3 (1)	2.9 (1)	54
*B. turdi*		0.3 (3)	50 (1)	0	0	0	0	0	3
*B. valaisiana*		1.9 (20)	0	0	0	0	0	0	15
*B.* spp.		0.1 (1)	0	0	0	0	0	0	1
*B. miyamotoi*		1.1 (11)	0	0	0	0	0	0	10
Genus *Ehrlichia*		1.4 (15)	0	0	0	0	0	0	12
*E. canis*		0.1 (1)	0	0	0	0	0	0	1
*close to E. chaffeensis*		0.5 (5)	0	0	0	0	0	0	5
*E.* spp.		0.9 (9)	0	0	0	0	0	0	9
Genus *Rickettsia*		6.5 (68)	0	0	18.2 (2)	6.7 (1)	0	32.3 (11)	59
*R. aeschlimannii*		0.4 (4)	0	0	0	0	0	0	4
*R. helvetica*		4.1 (43)	0	0	0	0	0	5.9 (2)	36
*R. slovaca*		0.1 (1)	0	0	0	0	0	26.5 (9)	2
*R.* spp.		1.9 (20)	0	0	18.2 (2)	6.7 (1)	0	0	21
Total larvae		1039	2	2	11	15	3	34	
Total birds									442

*^a^* Bbsl: Borrelia burgdorferi sensu lato, *^b^* Bbss: Borrelia burgdorferi sensu stricto.

**Table 3 pathogens-11-00946-t003:** List of primers used for confirmation using nested and conventional PCR.

Pathogen Genus	Target Gene	Primer Name	Sequence (5’-3’)	Amplicon Size (bp)	T	Reference
*Borrelia* spp.	*FlaB*	FlaB280F FlaRL flaB_737F FlaLL	GCAGTTCARTCAGGTAACGG GCAATCATAGCCATTGCAGATTGT GCATCAACTGTRGTTGTAACATTAACAGG ACATATTCAGATGCAGACAGAGGT	645 407	55 59	[95]
*Anaplasma* spp./*Ehrlichia* spp.	*16S rRNA*	EHR1 F EHR2 R EHR3 F EHR2 R	GAACGAACGCTGGCGGCAAGC AGTA(T/C)CG(A/G)ACCAGATAGCCGC TGCATAGGAATCTACCTAGTAG AGTA(T/C)CG(A/G)ACCAGATAGCCGC	693 629	60 55	[96]
*Rickettsia* spp.	*gltA*	Rsfg877 Rsfg1258	GGG GGC CTG CTC ACG GCG G ATT GCA AAA AGT ACA GTG AAC A-	381	56	[97]
*Babesia* spp.	*18S rRNA*	BTH 18S 1st F BTH 18S 1st R BTH 18S 2nd F BTH 18S 2nd R	GTGAAACTGCGAATGGCTCATTAC AAGTGATAAGGTTCACAAAACTTCCC GGCTCATTACAACAGTTATAGTTTATTTG CGGTCCGAATAATTCACCGGAT	1500	58	[98]
*B. miyamotoi*	*IGS*	Bospp-IGS-F Bospp-IGS-R Bospp-IGS-Fi Bospp-IGS-Ri	GTATGTTTAGTGAGGGGGGTG GGATCATAGCTCAGGTGGTTAG AGGGGGGTGAAGTCGTAACAAG GTCTGATAAACCTGAGGTCGGA	1007 388–685	56 58	[99]
Tick species	*COI*	HCO2198 LCO1490	TAA ACT TCA GGG TGA CCA AAA AAT CA GGT CAA CAA ATC ATA AAG ATA TTG G	710	48	[15]

F: forward; R: reverse; bp: base pairs; T: hybridisation temperature.

## Data Availability

Not applicable.

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
