# Peer review of "Diversity of Tick-Borne Pathogens in Tick Larvae Feeding on Breeding Birds in France"

_pathogens, 2022, doi:10.3390/pathogens11080946_

Round 1

Reviewer 1 Report

The Authors of manuscript titled “Diversity of tick-borne pathogens in tick larvae feeding on reproductive birds in France” submitted valuable results of study on pathogens detected in ticks found on birds during their breading season. Unfortunately, manuscript contains some issues that should be improved.

First of all, the Authors should consider modification of the manuscript title, as in my opinion “reproductive birds” should be replaced with “birds breeding”, “birds that reproduce” or another but more grammatically appropriate.

Secondarily, word “parasite” e.g. in line 355 should be replaced with name of particular systematic group of parasite species, i.e. protozoa. Of course, mentioned species of protozoa described by the Authors are parasites but as particular systematic name is known it should be used then.

Additionally, “p-value” should be replaced only with “p”. Information about value of p considered as statistically significant should be given in Methods and not repeated many times in Results. In Methods – 4.1 – percentage values of birds captured in particular months should be moved to Results – unless it was planned by the Authors to achieve these values during study – then it should be explained. Moreover, in L332-334 sentence should be edited or partially moved to the Results section. For now, this sentence contains both  - methods and results at once. The Authors should also redraft conclusion section to make it more perspicuous.

I hope I can have a question to Authors, as I did not found this information in the manuscript. I wonder if there was any significant difference among birds species, prevalence of tick infestation, pathogens prevalence year to year? Were these matters checked? I think that even there were no significant differences this information  should be added to the manuscript.

In conclusion, I believe that after improvement of mentioned issues, manuscript submitted by Amalia Rataud et al. should be considered for publication.

Reviewer 2 Report

This manuscript presents the results of two year monitoring of ticks collected on local breeding birds in France. Collected on their different life stages ticks were subjected for the tick-borne-pathogens analysis what let authors to point out the main hosts among trapped birds as the reservoir of set of different TBP. While the manuscript undoubtedly has a significant medical and veterinary value, it also shows the importance of the existed Constant ringing effort sites across the country where such kind of birds' parasites screening can be arranged. 

Generally, I found the article is well written but a few remarks I would raise. First one is a more discussion matter. While authors did not check birds' tissue and blood samples for pathogen agents, though they claim the prevalence of certain parasites among them based on PCR analysis of blood engorged ticks larvae (lines 337-339). I am pretty sure the authors see the difference and the usage of “rate of exposed birds” referring to number of birds which have been exposed to the potential risk of infection with infected ticks' larvae is more precisely described the situation and well avoid misinterpretation. Also, I wander why the authors use "larva infection rate" instead of prevalence, while prevalence (in my humble opinion) fits here well. Is it just because of the larvae were used for analysis only without nymphs and adult ticks? Please, clarify.

I would also suggest removing data of adult ticks and their nymphs from the manuscript as authors concentrate on information mostly obtained from larvae and the title of the manuscript is right about this.

Minor marks.

175, 206, 381 Latin name of species should be written in full in the beginning of a sentence.

322-324 This information would be better to move to Results.

Figure 3 (B) is quite a challenge to understand and moreover it does not give much of info as many bird species are presented in small numbers only.
